# Application of Human Brain Organoids—Opportunities and Challenges in Modeling Human Brain Development and Neurodevelopmental Diseases

**DOI:** 10.3390/ijms241512528

**Published:** 2023-08-07

**Authors:** Soo-hyun Kim, Mi-Yoon Chang

**Affiliations:** 1Department of Biomedical Science, Graduate School of Biomedical Science and Engineering, Seoul 04763, Republic of Korea; ksh93522@naver.com; 2Biomedical Research Institute, Hanyang University, Seoul 04763, Republic of Korea; 3Department of Premedicine, College of Medicine, Hanyang University, Seoul 04763, Republic of Korea; 4Hanyang Institute of Bioscience and Biotechnology, Hanyang University, Seoul 04763, Republic of Korea

**Keywords:** brain organoid, neurological disease, disease modeling, cell therapy, neural stem cell

## Abstract

Brain organoids are three-dimensional (3D) structures derived from human pluripotent stem cells (hPSCs) that reflect early brain organization. These organoids contain different cell types, including neurons and glia, similar to those found in the human brain. Human brain organoids provide unique opportunities to model features of human brain development that are not well-reflected in animal models. Compared with traditional cell cultures and animal models, brain organoids offer a more accurate representation of human brain development and function, rendering them suitable models for neurodevelopmental diseases. In particular, brain organoids derived from patients’ cells have enabled researchers to study diseases at different stages and gain a better understanding of disease mechanisms. Multi-brain regional assembloids allow for the investigation of interactions between distinct brain regions while achieving a higher level of consistency in molecular and functional characterization. Although organoids possess promising features, their usefulness is limited by several unresolved constraints, including cellular stress, hypoxia, necrosis, a lack of high-fidelity cell types, limited maturation, and circuit formation. In this review, we discuss studies to overcome the natural limitations of brain organoids, emphasizing the importance of combinations of all neural cell types, such as glia (astrocyte, oligodendrocytes, and microglia) and vascular cells. Additionally, considering the similarity of organoids to the developing brain, regionally patterned brain organoid-derived neural stem cells (NSCs) could serve as a scalable source for cell replacement therapy. We highlight the potential application of brain organoid-derived cells in disease cell therapy within this field.

## 1. Introduction

Research on the human brain and neurological diseases has primarily relied on post-mortem brain specimens and animal models. Specifically, post-mortem brain studies have allowed researchers to directly examine brain tissue in great detail [1,2,3,4]. However, this approach is limited to examining brains from deceased individuals; therefore, obtaining brain tissue from individuals with specific disorders or at certain developmental stages may not always be possible. To overcome the limited availability of human brain tissues, animal models have played a critical role in neuroscience research, providing insights into the genetic origins of certain diseases and enhancing the understanding of pathological processes at the cellular level [5,6,7,8]. However, due to the distinct complexity of the human brain, modeling human brain development and disease accurately using animal systems has been challenging [9]. This limited capability has resulted in many failures in central nervous system (CNS) drug development efforts [10]. In addition, the ethical implications of using animals in research need to be considered. Consequently, there has been a growing interest in developing physiologically relevant human brain research models derived from induced pluripotent stem cells (hiPSCs) [11]. By constructing hiPSCs from the patient’s tissues, it is possible to utilize patient-specific genetic information for disease modeling. This can be further utilized as a personalized medicine and precision medicine platform, enabling repeated testing through differentiation into various tissues. Human patient tissue-derived iPSCs can be utilized as a living biobank. By utilizing cells from hiPSCs, it is possible to create disease model platforms for elucidating reactions and interactions with external factors and harmful substances (such as infectious microorganisms, viruses, and toxic substances) that were previously impossible [12]. With the increased accessibility of gene correction technologies, such as CRISPR/Cas9, more sophisticated and novel attempts in personalized medicine research are becoming possible. 

However, the two-dimensional (2D) culture of these cells commonly used in the laboratory is not indicative of actual cell environments. Cultivating cells on a flat surface does not provide a comprehensive understanding of their growth and functionality within the human body, where they are surrounded by other cells in three dimensions (3D). The primary benefits of 3D cell culture involve enhanced interactions among cells, cell division, and morphology that closely resemble in vivo conditions [13]. The 3D shape more faithfully replicates the natural cellular environment, resulting in gene expression and morphology that better reflect the human body.

The current understanding of the human system and its dynamics has increased through the study of post-mortem and pathological samples, as well as animal models. Nevertheless, these approaches fall short in accurately predicting the clinical effectiveness of therapeutics, mainly due to differences in pharmacodynamics, and interspecies genetic or metabolic variations. Due to the limitations of modeling the in vivo complexity of the human CNS using animal models and 2D monolayer cultures, 3D brain organoids have been developed [14]. As shown in Figure 1, the brain organoid-based platform offers a high operational and application value because it overcomes the limitations of classical cell culture methods and allows for the easy utilization of existing efficient biochemical and cellular analysis techniques based on 2D culture [12,14,15]. Brain organoids derived from human patient iPSCs, such as fibroblast and blood-derived iPSCs, have been utilized to simulate a diverse range of neurological, developmental, and psychiatric disorders. By replicating disease phenotypes observed in patients, the modeling of microcephaly [16], Seckel syndrome [17], autism [18], Rett syndrome [19], and Miller–Dieker syndrome [20] in brain organoids has significantly deepened our comprehension of the pathobiology of these neurological disorders. Brain organoids that simulate brain development serve as excellent model systems for studying neurodevelopmental diseases [21,22,23,24]. However, these organoids have limitations, including cell immaturity, the absence of certain cell types (e.g., microglia, vascular cells), and the accumulation of cellular stress, which make them unsuitable for a comprehensive and accurate modeling of brain diseases that occur alongside cellular maturation [25,26]. Efforts have been made to address these limitations by incorporating non-existent cell populations (such as microglia and vascular cells) and creating structurally connected brain assembloid models [27,28,29,30]. In addition, dissociating ideal neuronal cells from brain organoids and combining them with decellularized extracellular matrix (ECM) technology to perform 3D disease modeling with these cells is possible [13,31]. Furthermore, organoids can be considered a cell supply source for disease cell therapy [32] rather than solely a disease model.

In this review, we explore the topic of brain organoids by presenting various studies. These studies highlight three key aspects: Firstly, their potential as alternatives to animal models or classical 2D culture systems in experiments and their significance as valuable models for studying brain developmental disorders. Secondly, we discuss the ongoing research efforts aimed at overcoming the limitations associated with brain organoids. And finally, we explore the applications of brain organoids, such as brain organoid-derived cells, as a scalable source for cell therapy.

## 2. Limitations and Potentials of Human Brain Organoids

### 2.1. Brain Organoids Reflect Brain Development, but with Clear Limitations 

Brain organoids consist of progenitors, neurons, and astrocytes that resemble those observed in the developing brain. They self-organize into cytoarchitectonic features similar to early brain organization [14,16,22,33,34]. To create brain organoids, embryoid bodies (EBs) derived from aggregates of human pluripotent stem cells/ human induced pluripotent cells (hPSCs/hiPSCs) are embedded into Matrigel^TM^ and cultured to promote tissue expansion and neural differentiation [35]. These organoids develop through self-organization without external interference [14]. During brain organoid development, similar to fetal brain development, organoids efficiently differentiate into diverse specific regional identities following an endogenous differentiation trajectory (Figure 2) [14,23,35,36,37]. Neuroepithelial cells give rise to radial glial cells, followed by other proliferating progenitors that resemble endogenous development at the transcriptional level [23,38]. The emergence of cells expressing markers of a superficial-layer identity is followed by those expressing deep-layer genes [16,33]. Astrocytes mature following neurons, mimicking the in vivo counterparts’ timescale [39,40].

Forebrain organoids consistently form cortical structures with distinct layers resembling the ventricular zone (VZ), the inner and outer subventricular zone (SVZ), and the cortical plate (CP) at molecular, cellular, and structural levels [16,23,47]. Midbrain organoids also exhibit features similar to early in vivo midbrains, particularly in developing neuroectoderm toward the floor plate [32,41,48]. Cortical organoids maintain the cortical plate’s structure in human fetal brain development, allowing research on laminar structure, layered neuron generation, temporal progression, migration, and maturation [23,47]. The enlarged outer subventricular zone in forebrain organoids offers unique advantages for studying human cortical development and developmental disorders [21,22,23,26]. However, cortical layer formation is a complex process involving the generation and migration of different types of neurons to form the six-layered structure of the mature cortex. Brain organoids do not fully capture the complexity and diversity of the human developing brain [22,25,26,49]. They are unable to fully model the interplay between intrinsic genetic programs and extrinsic cues, such as signaling molecules and cell–cell interactions, which are critical for the formation of cortical layers [22,25,28]. In addition, the brain organoids lack vascular cells and immune cells, such as microglia, which are crucial for maintaining a normal brain environment and promoting neuron maturation [28,40].

However, brain organoids offer a physiologically relevant in vitro 3D brain model for studying neurological development and disease processes unique to the human nervous system. Human patient-derived iPSCs have the advantage of maintaining the genetic characteristics of the patient. When utilizing these patient-derived hiPSCs in combination with genome-editing technologies, brain organoids offer a valuable system for investigating the development, diseases, and evolution of the human brain [50,51,52,53]. Brain organoids have important applications for studying human brain development and neurological disorders such as autism, schizophrenia, or brain defects caused by Zika virus infection [33,50,52].

### 2.2. Neurological Disorder Research Utilizing Brain Organoids 

The human brain is incredibly complex, and due to limited experimental models, inter-individual variability, and the need for ethical considerations, studying its developmental process is difficult. However, human brain organoids have emerged as a successful tool for modeling human brain development and neurodevelopmental diseases [22,26,52,54].

In 2016, the Zika virus, which spread through Central and South America, was identified as the cause of microcephaly in fetuses. However, a research model was not available to study the relationship between the virus and brain development. Brain organoids provided a breakthrough in elucidating the mechanisms through which Zika virus infection induces neural developmental impairments, including the death of neural stem cells (NSCs) and dysfunction in offspring [33]. Combined with hiPSCs or genome-editing technologies, human brain organoids have also helped to determine the pathological mechanisms in neurodevelopmental disorders [50,51,53,55]. For example, an organoid model of microcephaly associated with CDK5RAP2 mutation showed a significantly reduced size and smaller number of progenitors undergoing premature neurogenesis, similar to observations in patients [16]. Although animal models have been used to study aspects of neural development and pathological mechanisms, notable differences exist between animal brains and the human brain [32]. These discrepancies pose significant challenges in studying the development of the human CNS and related diseases [26,54]. Brain organoids can be used to study a specific disorder, such as Angelman syndrome (AS), a form of autism spectrum disorder caused by a loss-of-function mutation of the UBE3A gene [50]. In studies using AS mouse models, an impaired synaptic connectivity has been demonstrated with an imbalance between network excitation and inhibition and delayed neurodevelopmental processes. However, the underlying pathological mechanisms of seizures in AS have not been fully established. When characterizing the functional properties of cortical organoids derived from genome-edited UBE3A knockout human embryonic stem cells (hESCs) and AS-hiPSCs, an evolutionarily conserved channelopathy was identified that contributes to network dysfunction and hyperactivity in AS [50]. Another example is familial juvenile parkinsonism associated with loss-of-function mutations of DNAJC6. When using brain organoids, an explanation for the disease mechanism was provided [51]. In studies using knockout (KO) mice lacking Dnajc6, a high rate of unexplained early postnatal mortality and defects in synaptic recycling and Golgi–lysosomal trafficking were reported [56]. However, in these KO mouse studies, Parkinson’s disease (PD)-associated phenotypes, such as the loss of dopamine neurons and intraneuronal α-synuclein inclusions in the substantia nigra, were not observed. In contrast, human midbrain-like organoids carrying DNAJC6 mutations showed key PD pathologic features, including the degeneration of midbrain-type dopamine neurons, an increased intrinsic neuronal firing frequency, α-synuclein aggregation, and mitochondrial and lysosomal dysfunctions [51].

Numerous studies have consistently demonstrated that rodent models of PD and Alzheimer’s disease (AD) fail to replicate the same pathophysiology observed in human patients [57]. As a result, the brain organoid model is now regarded as a superior alternative, particularly for investigating the early phases of disease progression. Most neurodegenerative disorders typically manifest in adulthood [57]. However, in certain instances, neurodegenerative diseases such as AD and PD with early onset can be studied using hiPSC-derived brain organoid models [58]. For example, when brain organoids derived from patient hiPSCs carrying genetic risk factors like APP^dup^, APOE4, PSEN1, and PITRM1 associated with AD were examined, distinct early pathological characteristics, including amyloid aggregate accumulation, tau pathology, and neuronal cell death, were observed [58,59,60,61,62]. Similarly, in the PD model, patient hiPSC-derived midbrain organoids with genetic mutations of SNCA-A53T and LRRK2-G2019S exhibited dopaminergic neuronal loss and synaptic defects, faithfully recapitulating the pathological hallmarks and gene expression profiles seen in patients [63,64,65]. Among various in vitro models, hiPSC-derived brain organoids demonstrate a superior representation of these pathological features [61]. These findings underscore the advantages of 3D brain organoids over 2D cell cultures for modeling disease pathology. Even without patient hiPSCs, mechanistic studies can still be conducted using simple genetic backgrounds that incorporate PD or AD models [51,60,63,66,67] because the advantage of using genome editing technology is the ability to create isogenic organoid comparisons. This unique platform enables the modeling of early pathological features seen in degenerative diseases in humans. However, it should be noted that early onset AD or PD accounts for only a small percentage of all cases, less than 5%. The immaturity of hiPSC-derived brain organoids remains a challenge that needs to be overcome. The current brain organoid models of neurodegenerative diseases do not fully reflect the key pathological factors of aging, and due to the lack of mature communication between different tissues and cells, additional pathological features that arise in the actual physiological brain environment are not present. Furthermore, the absence of immune cells poses another limitation, restricting the use of brain organoids in modeling inflammatory responses to toxic or pathological substances, as well as age-related inflammation.

Brain organoids serve as valuable human in vitro models for studying neurological diseases, particularly those that cannot be fully evaluated using animal models.

### 2.3. Research Areas to Overcome Developmental Limitations of Brain Organoids

Brain organoids develop following an endogenous differentiation trajectory at the transcriptional level. Although organoids generally exhibit cell types that broadly reflect the expression profiles of human neural cells, their specification programs are impaired. In cortical brain development, a maturation process of progenitors, the areal specification of neurons, and the generation of diverse cell subtypes occur. However, cortical organoids fail to display appropriate cell-type maturation and distinct cellular subtype identities. Single-cell transcriptomics can be utilized to identify and validate molecular cell signatures across cortical areas in the developing brain and in cortical organoids [25,36,68]. A comparison of the human cortex and organoids has shown that organoids have 0.45-fold fewer cells expressing the outer radial glia marker HOPX and 2.5-fold more positive cells expressing the NSC marker SOX2 [25,36,68].

Primary brain neurons, in contrast to cortical organoids, exhibit a low expression of radial glia markers, and radial glia do not express neuronal markers [25]. In addition, genes involved in neuronal development and projection pattern specification, such as MEF2C and SATB2, are significantly upregulated only in primary cells [25,36]. Organoids exhibit a chronic ectopic expression of stress-associated genes across all cell types [49]. Cellular stress appears to be a broader feature of in vitro culture, as evidenced by the enrichment of glycolysis gene PGK1 and endoplasmic reticulum stress genes ARCN1 and GORASP2 at the gene and protein level in organoids [25,49]. Chronic cellular stress, which is not characteristic of normal neural development, can disrupt developmental processes such as fate specification, maturation, morphology, or connectivity. Numerous potential sources of metabolic dysregulation, such as hypo-oxygenation, an inadequate supply of essential nutrients and sugar levels, or the absence of crucial cell types and structures like vasculature or cerebrospinal fluid flow, may contribute to increased stress levels in organoids. These ectopically activated cellular stress pathways impair cell-type specification and hinder the complete recapitulation of the differentiation trajectories of cortical neurons in vivo.

Organoids, being 3D masses grown over time in culture media, can display hypoxia and necrosis within the mass. However, cultivating flat disc-shaped organoids has resulted in the formation of more extended axon bundles [69,70]. By implementing air–liquid interface culture techniques in cerebral organoids [69], neuronal survival and axon outgrowth have shown significant improvement. Through single-cell RNA sequencing, diverse cortical neuronal identities have been identified, while retrograde tracing has confirmed the alignment of tract morphologies with specific molecular identities. The utilization of disc slice culture has overcome diffusion limitations [70], ensuring the prevention of interior cell death and facilitating a sustained organoid growth during long-term cultivation. Consequently, the resulting organoids accurately replicate advanced developmental characteristics of the human cortex, including the formation of distinct cortical layers [70]. By effectively controlling fluidic flow and improving nutrient and oxygen delivery to cells, microfluidic devices have the potential to minimize the hypoxic core in brain organoids. This enhancement could result in a greater maturity and an improved functionality compared to traditional cell culture conditions [15]. Together, the use of human brain extracellular matrix (ECM) and periodic flow could improve cell survival [71]. Moreover, employing a more multidisciplinary engineered strategy to generate human brain assembloids holds the potential as a future technology [72]. Brain organoid-on-a-chip technology offers a promising approach to replicate brain wrinkling and folding mechanisms that occur during brain development [73]. 

The absence of a vascular system or certain cell types such as glia can contribute to hypoxia and stress in organoids [28]. And the utility of brain organoids is hindered by the absence of those cell types, which play a crucial role in regulating neurogenesis and brain disorders. Efforts have been made to include vascular cells or microglia, which originate from different lineages [27,30,74]. The induction of vasculature-like channels within organoids, achieved through human umbilical vein endothelial cell transplantation [74] or the ectopic expression of endothelial genes (ETV2) [27], decreases cell death and improves neurogenesis. Brain organoids were combined with vascular organoids to create vascularized brain organoids [75]. The fused organoids not only featured functional blood–brain barrier-like structures but also demonstrated an increased abundance of neural progenitors. The cerebral microvasculature secretes brain developmental cytokines like BDNF (brain-derived neurotrophic factor) [76]. Brain organoids have shown enhanced development when co-cultured with a vasculature system, which can support their growth and maturation through the secretion of BDNF [77]. These findings suggest that the presence of vessels contributes to the regulation of neural development [27,74,75,77]. Gliogenesis follows neurogenesis in organoids, although astrocytes are often produced; however, oligodendrocyte precursor cells (OPCs) and mature oligodendrocytes are rarely observed [23,40]. Microglia, the immune cells of mesodermal origin residing in the CNS, are challenging to locate within brain organoids [26]. Microglia have phagocytic potential for pathogens and unused synapses, playing a role in inflammation and circuit refinement [78]. In brain organoid models, microglia are an indispensable cellular component [78]. Co-culturing human microglia with brain organoids allows integrated microglia to mature, ramify, and to respond to injuries similarly to in vivo microglia [24,29,79]. These similarities to their in vivo counterparts indicate that the microenvironment of brain organoids maintains the homeostatic state of microglia [29]. The transplantation of microglia into brain organoids triggered transcriptional alterations, alleviated cell stress, and reduced the expression of genes related to interferon response. Furthermore, microglia enhanced the synchronization and frequency of oscillatory bursts in the brain organoids, promoting the maturation of neural networks [29]. These observations underscore the significant role of microglia in neural development [79]. In numerous studies, attempts have been made to address the inherent challenges of organoids, but limitations remain, such as an indistinct batch reliability and a limited recapitulation of mature development.

### 2.4. Neural Circuit Research Utilizing Fusion Assembloids of Regional Brain Organoids

Neurons in the cerebral cortex establish connections with other regions of the brain through white matter tracts that are not present in organoids. Consequently, organoids cannot be used to study how the cortex communicates with other brain regions. Although different brain regions closely communicate with each other under normal conditions, achieving this level of interaction in spontaneously developed organoids is not feasible. These limitations restrict their usefulness in neural connectivity research. Consequently, despite the crucial role that interconnections between regional brain tissues play in normal brain function, extensive modeling using this approach is not possible. For example, disordered corticostriatal connectivity has been implicated in several neurodevelopmental, neuropsychiatric, and movement disorders—schizophrenia, autism, amyotrophic lateral sclerosis, Huntington’s and Parkinson’s diseases, and major depression [42,80]. Thalamic dysfunction in projections between the thalamus and cortex has been associated with neurodevelopmental disorders, including autism, schizophrenia, and epilepsy [43]. The loss of connectivity in the nigrostriatal pathway, a bilateral dopaminergic pathway in the brain, is primarily involved in PD [81]. Cortical–hypothalamic circuits mediate stress integration [82], and the hippocampal–hypothalamic circuit serves as the control center of the hypothalamic–pituitary–adrenal (HPA) axis involved in anxiety disorder, bipolar disorder, insomnia, ADHD, and alcoholism [83].

To improve the modeling of inter-regional interactions, several groups have developed separate organoids resembling specific brain regions [84]. Diverse regionally patterned protocols utilize small molecules with recombinant cytokines to induce developmental signaling and promote specific regional identity (Figure 2). Various protocols for generating region-specific brain organoids have been established, including those patterning the dorsal forebrain [14,18,85], ventral forebrain [43,86,87], retina [88], hippocampus [44], thalamus [43], hypothalamus [33], midbrain [32,41,67], pituitary gland [89], cerebellum [45], and choroid plexus [46]. These different regional-patterned organoids can be generated and then fused together into a multi-region assembloid [39,43,86,87]. Studies have been conducted on thalamocortical [43], corticostriatal [42], hypothalamic–pituitary [90], and corticospinal–muscle [84] interactions. Corticostriatal and hypothalamic–pituitary assembloids have been shown in vivo-like axonal projections and the accelerated maturation of striatal neurons and pituitary, respectively. Recently, 3D cortical organoids were transplanted into the primary somatosensory cortex (S1) of early developmental stage rats (postnatal days 3–7) to create an in vivo neural circuit research platform [91]. Neurons from transplanted organoids extended axonal projections into the rat brain, with thalamocortical and corticocortical inputs evoking sensory responses that drive reward-seeking behaviors. This integrated organoid in vivo platform represents a powerful resource to complement in vitro studies of neural circuitry [91]. However, further work is needed to reliably assess connectivity because the connectivity between neurons in organoids is often incomplete and not as robust as in the mature human cortex. This limits their usefulness for studying the function of circuits and neural processing compared with actual models.

### 2.5. Applications—Brain Organoid-Derived NSCs for Cell Therapy

In addition to their application in investigating tissue development and human disease modeling, grafted organoids of intestinal, hepatic, nephric, or pancreatic tissue can serve therapeutic functions as functional units, such as liver buds, kidney nephrons, and pancreatic islets, after transplantation [92,93,94,95,96]. However, transplanting brain organoids is unlikely to be applied to brain disorder treatments due to the difficulty of grafting them into deep regions of the brain without damaging the host brain tissue. Furthermore, to obtain therapeutic outcomes of transplanted brain organoids, self-organized structures within the graft must establish new neural networks that interact precisely with the host brain. The brain organoids contain tissue-resident cell types and reflect features of early tissue organization; specifically, these have a scientific advantage given the lack of accessibility of developing human brain tissue. Instead of using brain organoids, the use of NSCs/precursor cells isolated from specifically patterned brain organoids has been a challenge [32] for cell replacement therapy (Figure 3). The cellular composition of cortical and midbrain organoids has been studied using both bulk and single-cell RNA sequencing approaches [22,32,36,41,48]. At early time points, cortical organoids reproduce aspects of human neuroepithelium physiology and contain progenitor cell clusters [23,97] resembling those found in primary human fetal samples [98]. Similar findings have been observed in midbrain organoids, which exhibit transcriptomic profiles close to those of the human fetal midbrain [41,48]. NSCs derived from organoids also exhibit characteristics closer to midbrain organoids and prenatal midbrain and maintain physiological traits after isolation from the organoid and multiple rounds of cell proliferation outside the organoid environment [32] (Figure 3). Compared with 2D in vitro cultured NSCs, midbrain organoid-NSC-derived midbrain dopaminergic neurons show improved synaptic maturity, functionality, resistance to toxic insults, and potent therapeutic outcomes that are reproducible in PD animal models [32]. Consequently, NSCs formed in organoid environments provide a better cell source for PD therapies. Therapeutic properties of NSCs have also been confirmed in human neurodegenerative diseases (NDs) such as ischemic stroke, amyotrophic lateral sclerosis (ALS), and multiple sclerosis (MS) [99,100,101]. In NDs, specific subsets of neurons, such as dopaminergic and cholinergic neurons or motor neurons, progressively degenerate, resulting in a specific pattern of nervous system dysfunction [101]. Specifically patterned organoids simulating different human brain regions could be effectively used for cell replacement therapy for NDs as well as for other approaches with NSC transplantation. For example, mouse hypothalamic NSCs were reportedly involved in regeneration, aging, and metabolic functions in aged animal models [102,103,104]. By patterning exact hypothalamic organoids, anti-aging and metabolic studies after transplantation into aged models could be investigated.

The use of organoid-derived cells in disease cell therapy is a potential application of this rapidly growing field. Organoid-derived cell therapy shows promise for treating a wide range of diseases, including cancer, NDs, and anti-aging. Recent research has also shown that tissue-specific organoid-derived cells can be used to repair damaged cardiac tissue [93,105] and restore healthy liver tissue in patients with liver failure [92,105]. However, the use of brain organoid-derived NSCs in therapy is still in its early stages and more research is needed to fully understand their potential and limitations.

The use of animals in disease or research models is gradually decreasing, as they involve limitations, especially in drug development, due to dissimilarities from humans. The drug discovery process faces a significant bottleneck issue due to challenges in extrapolating results from model systems to humans. Organoid technology holds the potential for a revolutionary breakthrough, offering a bridge between preclinical and clinical trials. This advancement could allow for more reliable testing models in precision medicine and drug discovery for neurological disorders [12,106].

## 3. Discussion

Human ESCs/iPSCs-based brain organoids provide a valuable platform for modeling features of normal and pathological development. Brain organoids can also be utilized as a disease modeling platform to investigate responses and interactions with external factors and harmful substances (such as infectious microorganisms, viruses, and toxic substances) that were previously difficult to study using animal models. However, the aforementioned limitations regarding research utilizing brain organoids remain significant hurdles that still need to be overcome, including the challenge of achieving experimental reproducibility during organoid generation. To address these limitations, improved technologies were adopted, utilizing different culture methods such as flat disc, liquid air culture, and a spinning bioreactor to enhance gas and nutrient exchange. Also, the addition of essential cell types (e.g., microglia, vascular cells) that are not naturally present, or the creation of assembloids using multiple organoids, can lead to a closer mimicry of actual brain tissue and enhance neuronal maturation. When utilizing these approaches, an advance modeling of neurodevelopment and neurodevelopmental disorders is possible. However, for the most crucial models of CNS degenerative disorders, observing the dynamics among all cells within the brain structure is important. Furthermore, because these degenerative disorders often occur in conjunction with aging, the current brain organoid technology that cannot reflect such characteristics has obvious limitations. The next step of utilizing brain organoid is to conduct various disease modeling studies using new cells derived from the organoid. These cells could also potentially lead to cell therapy for degenerative diseases. Because brain organoids also exhibit similarities to actual fetal brains through 3D cultivation, utilizing them as a more physiologically relevant cellular supply source, similar to other tissue-derived organoids (e.g., liver, cardiac, intestine), is a promising approach. 

Despite their limitations, no model is perfect, and brain organoids remain the only viable alternative to animal disease models. Utilizing brain organoids as a platform for various research and disease studies represents a promising future avenue of stem cell research.

## Figures and Tables

**Figure 1 ijms-24-12528-f001:**
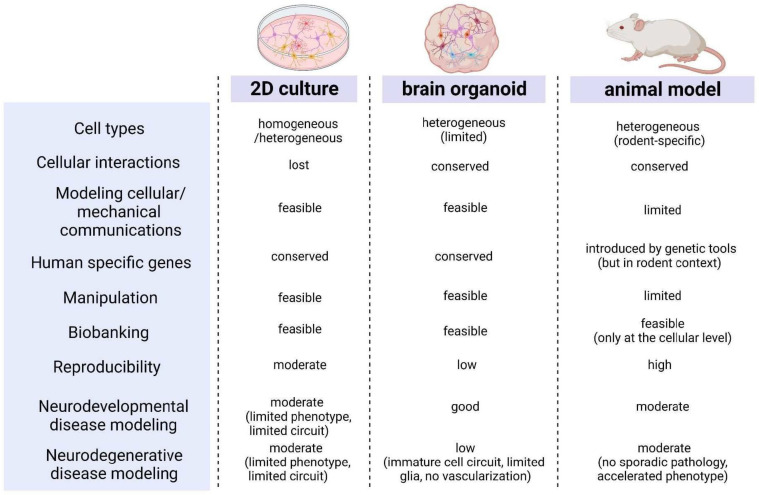
Comparison of characteristics among two-dimensional (2D) cell culture, three-dimensional (3D) brain organoid models, and animal models. Organoids have advantages compared to 2D cultures and animal models, and can be a practical platform for modeling diseases with conserved human-specific genes and cellular interactions that enable biobanking and cellular manipulations.

**Figure 2 ijms-24-12528-f002:**
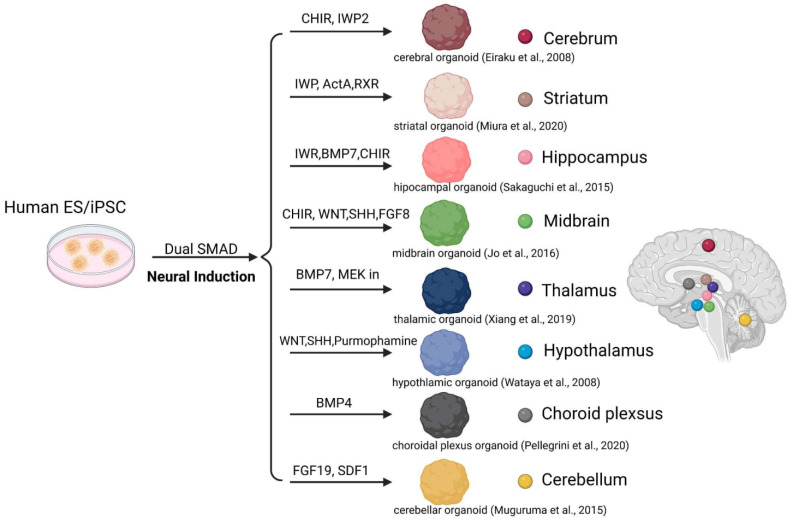
The production of brain region-specific organoids through patterned differentiation using an appropriate combination of cytokines/small molecules derived from human embryonic stem cells (hESCs). [14,41,42,43,44,45,46].

**Figure 3 ijms-24-12528-f003:**
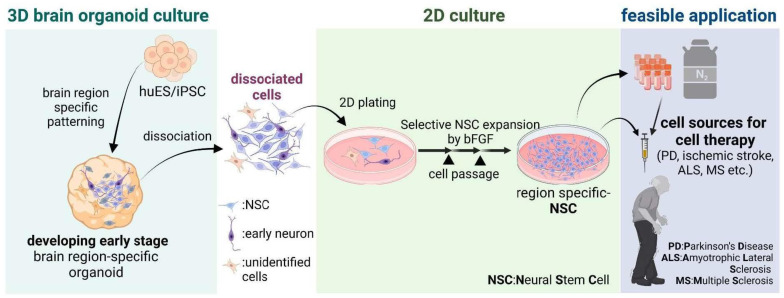
Based on the characteristics of human brain organoids that mimic the molecular/cellular features of fetal brains, specific organoids corresponding to different brain regions are created. From these organoids, region-specific neural stem cells (NSCs) are dissociated and utilized as a cell source for neurodegenerative diseases such as Parkinson’s disease (PD), amyotrophic lateral sclerosis (ALS), and multiple sclerosis (MS).

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
