# Peer review of "Application of Human Brain Organoids—Opportunities and Challenges in Modeling Human Brain Development and Neurodevelopmental Diseases"

_ijms, 2023, doi:10.3390/ijms241512528_

Round 1

Reviewer 1 Report

Comments and Suggestions for Authors

Application of human brain organoids: a system for human brain biology and disease

Soo-hyun Kim and Mi-Yoon Chang

In this manuscript, the authors present a review on human brain organoids derived from human pluripotent stem cells (hPSCs) and focus their attention on the advantages and limitations of using these organoids for the study of brain development and neurological disorders. The authors make a comparison between organoids, 2D cultures and animal models, concluding that the use of organoids is a more valid method to study a broad spectrum of neurological disorders. Finally, the authors identify the limitations of hPSCs-derived brain organoids and discuss innovative applications to make the most of their enormous potential, such as the creation of assembloids to study brain connections in the context of brain development research, or even the use of NSCs isolated from organoids for cell replacement therapy. Brain organoid research is very fascinating, but it is still in its infancy and there is an urgent need for wide-ranging knowledge to fully understand the great potential of brain organoids in basic and clinical research. Below are my suggestions for improving the work written by Soo-hyun Kim and Mi-Yoon Chang.

Comments and suggestion for authors:

  1. I suggest authors to include more references, and to add more articles than reviews.
  2. Authors should elaborate more on the concepts in the abstract section.
  3. Ref. number 12 at line 71 and ref number 17 at line 77 are out of context, please remove them.
  4. Line 65: Please add “human” prior to “CNS” and “animal models and” prior to “2D-monolayer cultures”.
  5. I suggest authors to comment Figure 1 in the text, where did you get the information to create it? Please reformulate also figure 1 legend.
  6. Lines 85-88: please reformulate better this concept.
  7. Line 96: Based on the meaning of what is stated here, I suggest substituting “normal brain development” with “foetal brain development”.
  8. Line 136: Delete “(Fig. 1)” at the end of the sentence.
  9. Line 215: Please substitute “neuronal maturation” with “neuronal development”.
  10. Lines 216-219:  Here the authors discuss the well-known issue of cellular stress in organoids, I suggest stressing this concept more.
  11. Lines 222-223: Please explain better why there is presence of hypoxia and necrosis in organoids.
  12. Paragraph 1.3: Here, the authors describe some research areas dealing with overcoming the limitations of brain organoids but in my opinion they forgot to mention the following: the use of dynamic culture (orbital shaker, spinning flasks and bioreactors) to improve culture condition and to facilitate nutrients exchange, the use of microfluidics to obtain organs-on-a-chip platforms to better mimic the in-vivo condition, and the use of 3D bioprinting to reduce variability. I suggest authors giving a few hints of these topics.
  13. Lines 270-272: “For example, disordered cortico-striatal connectivity has been implicated in several neurodevelopmental, neuropsychiatric, and movement disorders”. Please list which disorders.
  14. In section 1.5 authors discuss the use of brain organoids in cell therapy, I suggest also discussing their use in drug development.
  15. Line 373: please delete “our”.
  16. I have found some typos in the text, please correct them.

Comments on the Quality of English Language

Minor editing of English language

Author Response

Journal: IJMS (International Journal of Molecular Sciences)

Section: Biochemistry

Research Topics: Organoids: The New 3D-Frontier to Model Different Diseases In-Vitro

July 25. 2023.

Dear, Reviewer

RE:  Revision of the manuscript ijms-2497659

Thank you for your careful evaluation of our paper and for the additional opportunity to revise it for publication in IJMS.

Based on your suggestions and comments, we revised our paper as follows.

Please find the revised manuscript enclosed together with our point-by-point responses to the comments made by the reviewer.

We are very appreciative of your support of our paper publication.

Mi-Yoon Chang, PhD

Department of Premedicine

College of Medicine, Hanyang University

133-791, Seoul, KOREA

Tel: 82-2-2220-0620

Fax: 82-2-2220-2422

E-mail: mychang@hanyang.ac.kr

Reviewer 1:

In this manuscript, the authors present a review on human brain organoids derived from human pluripotent stem cells (hPSCs) and focus their attention on the advantages and limitations of using these organoids for the study of brain development and neurological disorders. The authors make a comparison between organoids, 2D cultures and animal models, concluding that the use of organoids is a more valid method to study a broad spectrum of neurological disorders. Finally, the authors identify the limitations of hPSCs-derived brain organoids and discuss innovative applications to make the most of their enormous potential, such as the creation of assembloids to study brain connections in the context of brain development research, or even the use of NSCs isolated from organoids for cell replacement therapy. Brain organoid research is very fascinating, but it is still in its infancy and there is an urgent need for wide-ranging knowledge to fully understand the great potential of brain organoids in basic and clinical research. Below are my suggestions for improving the work written by Soo-hyun Kim and Mi-Yoon Chang.

Comments and suggestion for authors:

  1. I suggest authors to include more references, and to add more articles than reviews.

--→ We have added twelve more article references exactly where they should be placed.

  1. Authors should elaborate more on the concepts in the abstract section.

--→ The abstract has been revised to accurately convey our intended meaning. The revised part is highlighted in red. (P.2 line 11~13, line 16~22 in revised version)

  1. Ref. number 12 at line 71 and ref number 17 at line 77 are out of context, please remove them.

--→ As reviewer’s comment, we noticed that the reference was out of context. Therefore, we removed the ref. 12 from line 71 and ref. 17 from line 77 (original version). – (P.4 line 2, and lines 5 in revised version)

  1. Line 65: Please add “human” prior to “CNS” and “animal models and” prior to “2D-monolayer cultures”.

--→ As reviewer’s suggestions, we added “human” before “CNS” and “animal models and” before “2D-monolayer cultures”. (P.3. line37~38 in revised version)

  1. I suggest authors to comment Figure 1 in the text, where did you get the information to create it? Please reformulate also figure 1 legend.

--→ As reviewer’s suggestions, we added a paragraph explaining the background for figure 1 (P.3 line33~36 in revised version). Additionally, we revised the legend for figure 1 (P.13 line 4~7 in revised version).

  1. Lines 85-88: please reformulate better this concept.

--→ We reformulated the line 85~88 (original version) (P.4 line12~15 in revised version).

  1. Line 96: Based on the meaning of what is stated here, I suggest substituting “normal brain development” with “foetal brain development”.

--→ As reviewer’s comment, we changed “normal brain” into “fetal brain” (P.5 line 9 in revised version).

  1. Line 136: Delete “(Fig. 1)” at the end of the sentence.

--→ We deleted (P.6 line 5 in revised version)..  

  1. Line 215: Please substitute “neuronal maturation” with “neuronal development”.

--→ As reviewer’s comment, we changed “neuronal maturation” into “neuronal development” (P.8 line 1 in revised version).

  1. Lines 216-219: Here the authors discuss the well-known issue of cellular stress in organoids, I suggest stressing this concept more.

--→ As reviewer’s suggestions, we emphasized the cellular stress of brain organoids (P.8 line 2~3, line6~11 in revised version).

  1. Lines 222-223: Please explain better why there is presence of hypoxia and necrosis in organoids.

--→ As reviewer’s suggestions, we added details about stress factors including hypoxia and necrosis (P.8 line 2~3, line6~11 in revised version).

  1. Paragraph 1.3: Here, the authors describe some research areas dealing with overcoming the limitations of brain organoids but in my opinion they forgot to mention the following: the use of dynamic culture (orbital shaker, spinning flasks and bioreactors) to improve culture condition and to facilitate nutrients exchange, the use of microfluidics to obtain organs-on-a-chip platforms to better mimic the in-vivo condition, and the use of 3D bioprinting to reduce variability. I suggest authors giving a few hints of these topics.

--→ As reviewer’s suggestions, we have already mentioned the use of dynamic culture conditions in this section (ref. 29,31,59, 60). Additionally, we have provided further details on how 3D microfluidic technology offers a promising approach to replicate brain environments (P.8 line 24~29 in revised version).

  1. Lines 270-272: “For example, disordered cortico-striatal connectivity has been implicated in several neurodevelopmental, neuropsychiatric, and movement disorders”. Please list which disorders.

--→ As reviewer’s comment, we added a list of disorders related to disordered cortico-striatal connectivity (P.9 line 26~27 in revised version).

  1. In section 1.5 authors discuss the use of brain organoids in cell therapy, I suggest also discussing their use in drug development.

--→ As reviewer’s suggestions, we changed the subtitle of 2.5. (P.10 line 15 in revised version) and added one paragraph explaining the importance of drug development with brain organoids (P.11 line 15~21 in revised version).

  1. Line 373: please delete “our”.

--→ As reviewer’s comment, we deleted “our”.

  1. I have found some typos in the text, please correct them.

--→ We checked the manuscript thoroughly and corrected the typo again.

Reviewer 2 Report

Comments and Suggestions for Authors

The human brain organoids, in vitro culture and their application as models for basic research and biomedical use is a groundbreaking technique gaining the increasing attention and interest. Therefore, the topic is really relevant and worth reviewing. However, there are some excellent recent reviews on the brain organoids in the literature, e.g. Eichmüller, O.L., Knoblich, J.A. Human cerebral organoids - a new tool for clinical neurology research. Nat Rev Neurol 18, 661–680 (2022). The purpose of this short review remains unclear, and the selected title is overly broad, while the presented review is not. Therefore, I suggest some corrections:

Comment 1:  the Authors should consider either modifying the title to address a specific target area suitable for a short review article or expanding the manuscript.
Comment 2:  the abstract should be corrected, I suggest to add a statement highlighting the significance and novelty of this review, as it is not currently evident from the text. Additionally, it would be beneficial to end the abstract with a concluding statement to provide the importance of this review in  covering the existing research gap.
Comment 3: The sections within manuscript are confusing – it seems like the Authors divided the whole article into 2 sections: Introduction and Discussion. Please, correct this.
Comment 4: The paper contains some errors and typos that sometimes make it difficult to understand. Therefore, the English revision is needed.
Comment 4: The figures presented are fine and well described with informative captions. 

Comments on the Quality of English Language

In general, English is fine but it could be improved and polished by native reviewer.

Author Response

Journal: IJMS (International Journal of Molecular Sciences)

Section: Biochemistry

Research Topics: Organoids: The New 3D-Frontier to Model Different Diseases In-Vitro

July 25. 2023.

Dear, Reviewer

RE:  Revision of the manuscript ijms-2497659

Thank you for your careful evaluation of our paper and for the additional opportunity to revise it for publication in IJMS.

Based on your suggestions and comments, we revised our paper as follows.

Please find the revised manuscript enclosed together with our point-by-point responses to the comments made by the reviewer.

We are very appreciative of your support of our paper publication.

Mi-Yoon Chang, PhD

Department of Premedicine

College of Medicine, Hanyang University

133-791, Seoul, KOREA

Tel: 82-2-2220-0620

Fax: 82-2-2220-2422

E-mail: mychang@hanyang.ac.kr

Reviewer 2:

The human brain organoids, in vitro culture and their application as models for basic research and biomedical use is a groundbreaking technique gaining the increasing attention and interest. Therefore, the topic is really relevant and worth reviewing. However, there are some excellent recent reviews on the brain organoids in the literature, e.g. Eichmüller, O.L., Knoblich, J.A. Human cerebral organoids - a new tool for clinical neurology research. Nat Rev Neurol 18, 661–680 (2022). The purpose of this short review remains unclear, and the selected title is overly broad, while the presented review is not. Therefore, I suggest some corrections:

Comment 1:  the Authors should consider either modifying the title to address a specific target area suitable for a short review article or expanding the manuscript.

--→ As reviewer’s comment, we changed the title to “Application of human brain organoids- opportunities and challenges in modeling human brain development and neurodevelopmental diseases”.

Comment 2:  the abstract should be corrected, I suggest to add a statement highlighting the significance and novelty of this review, as it is not currently evident from the text. Additionally, it would be beneficial to end the abstract with a concluding statement to provide the importance of this review in covering the existing research gap.

--→ As reviewer’s suggestions, we highlighted the significance of this Review and suitable concluding sentence in the ‘Abstract’ section (P.2 line 11~13, line 16~22 in revised version).

Comment 3: The sections within manuscript are confusing – it seems like the Authors divided the whole article into 2 sections: Introduction and Discussion. Please, correct this.

--→ We agree that the manuscript sections are confusing. We have divided it into three parts: 1. Introduction; 2. Limitations and Potentials of Human Brain Organoids; 3. Conclusion. Additionally, for ‘section 2’, we have further divided into 5 sub-sections.

Comment 4: The paper contains some errors and typos that sometimes make it difficult to understand. Therefore, the English revision is needed.

--→ We checked the manuscript thoroughly and corrected the typo again. For English revision, this revised text has been sent to the English editing company.

Comment 4: The figures presented are fine and well described with informative captions.

--→ Thanks for reviewer’s positive comment.

Round 2

Reviewer 1 Report

Comments and Suggestions for Authors

The authors have addressed all the requested revisions, and while the current work may not introduce significant innovations in the field, it remains informative and, from my perspective, can be accepted in its present state.

Comments on the Quality of English Language

Minor editing required

Author Response

Mr. Phitsanu Charoensuk

Assistant Editor

E-Mail: Phitsanu@mdpi.com

Journal: IJMS (International Journal of Molecular Sciences)

Section: Biochemistry

Research Topics: Organoids: The New 3D-Frontier to Model Different Diseases In-Vitro

Aug 2. 2023.

Dear, Mr. Phitsanu Charoensuk

Editor, International Journal of Molecular Sciences (IJMS)

RE:  Revision of the manuscript ijms-2497659

Thank you for your careful evaluation of our paper.

Based on the reviewer’s suggestions, we revised our paper as follows.

Please find the revised manuscript enclosed together with our point-by-point responses to the comments made by the reviewers.

We are very appreciative of your support of our paper publication.

Mi-Yoon Chang, PhD

Department of Premedicine

College of Medicine, Hanyang University

133-791, Seoul, KOREA

Tel: 82-2-2220-0620

Fax: 82-2-2220-2422

E-mail: mychang@hanyang.ac.kr

Comment 1. There is an improper and confuse use of hPSCs and iPSC in the manuscripts, please revised carefully. Below some examples and comments: Abstract: “Brain organoids are three-dimensional (3D) structures derived from human pluripotent 13 stem cells (hPSCs)...” Lines 49-51: “…human brain research models derived from induced pluripotent stem cells (iPSCs). By constructing iPSCs from the patient's tissues…” The authors mentioned the possibility to obtain iPSC from biopsy of the patients. The concept should be describe more deeply. The iPSC to generate brain organoids could derive both by biopsy or lymphocytes or fibroblasts from which it is possible to obtain iPSC with the advantage to maintain the genome of the patient. It is very difficult to obtain brain organoids from brain biopsy. Please, clarify also what is an iPSCs, as a review all the information has to be report.

→ In the “Abstract”, hPSCs refer to all human cell types with pluripotency capable of generating brain organoids. However, as pointed out by the reviewer in the “Introduction”, this definition could be slightly confusing, so we have made revisions. We added a sentence to provide further clarification and included additional references about hiPSC-derived brain organoid studies (line 1~6, P4, in revised version).

Line 108: “…derived from aggregates of human pluripotent stem cells (hPSCs)”, Generally the human brain organoids derived from iPSC because of the ethics problems to obtain human pluripotent stem cells.

→ The sentence in line 108 (original version) is a general explanation of the brain organoid generation methodology. We agree that, in response to the reviewer’s comment, we changed the hPSC into hPSC/iPSC in line 108 (original version).

Comment 2. Line 140: “…when combined with recent patient-derived hiPSC technologies”, please clarify better what are hiPSC technology and uniform hiPSC and iPSC.

→ As reviewer’s comment, we revised the sentences (lines 34~37, P 5 in revised version) and uniform iPSCs into hiPSCs.

Comment 3. Lines 148-150: “The current understanding of the human brain is mostly based on post-mortem…” the same concept has been repeated in the introduction (line 37-38 and lines 41-42). Please, edit the introduction of the paragraph 2.2

→ As reviewer’s comment, we have revised the introduction part of section 2.2. (lines 2~4, P 6 in revised version).

Comment 4. Lines 224-226: “Comparisons have show that organoids…” it is not clear with whom the comparisons are made. Please clarify.

→ As reviewer’s point, we have clarified the sentence to avoid confusion (lines 36~37, P 7 in revised version).

Comment 5. Lines 250: please, because microfluidic systems are an innovative technologies I think that is better go further in the discussion of the use of this device to culture brain organoids. Please refers to: Nat Commun 12, 4730 (2021). https://doi.org/10.1038/s41467-021-24775-5; Adv Mater 2023 35(14):e2210083 doi: 10.1002/adma.202210083.

→ As reviewer’s point, we have revised the manuscript and added important references (lines 28~30, P 8 in revised version).

Comment 6. Lines 270: “ These findings suggest that the presence of vessels contributes to the regulation of 270 neural development” the absence of vasculature limitation is an important problem for culturing organoids. In particular, it has been demonstrated that the absence of BBB seems to delay the brain organoids differentiation, please refers to: Biochem Biophys Res Commun 2022 626:30-37.doi: 10.1016/j.bbrc.2022.07.112.

→ As reviewer’s point, we have added the reference (line 1, P9 in revised version).

Comment 7. I think that the Paragraph 2.4 is redundant and could be summarized and moved at the end of the paragraph 2.1

→ Thanks for the review’s suggestion. However, nowadays, as scientists researching brain organoids realize their limitations, the importance of brain assembloids technology has become prominent for replicating the real brain environment. We intend to keep Section 2.4 as a separate section.

Comment 8. Paragraph 2.5 is untitled “APPLICATIONS”, however seems to be an extension of paragraph 2.2. I think that this paragraph could be delete and some part as lines 344-361 could be moved in the paragraph 2.2 and the rest of the paragraph could be re-elaborate and moved in the discussion.

→ Thank you for the reviewer's suggestion. We believe that including lines 344-361 (in the original version) is essential because they provide crucial evidence that regional-specific brain organoid-derived cells could be valuable for cell replacement therapy. These lines explain that, at early time points, cortical and midbrain organoids reproduce aspects of human neuroepithelium physiology and contain high populations of progenitor cell clusters. Numerous studies have demonstrated that fetal brain-derived neural stem cells are reliable cell sources for cell therapy. In comparison to other brain organoid-related reviews, our review will stand out, particularly in the last "Application" section, which we do not want to merge with other sections to maintain the strength of this review.
